# Essential Factors for a Healthy Microbiome: A Scoping Review

**DOI:** 10.3390/ijerph19148361

**Published:** 2022-07-08

**Authors:** Patricia Grace-Farfaglia, Heather Frazier, Maura Daly Iversen

**Affiliations:** 1Health Sciences, College of Health Professions, Sacred Heart University, Fairfield, CT 06825, USA; 2Department of Nutrition, School of Mathematics, Science and Engineering, University of the Incarnate Word, San Antonio, TX 78209, USA; hrfrazie@uiwtx.edu; 3Public Health and Physical Therapy and Human Movement Sciences, College of Health Professions, Sacred Heart University, Fairfield, CT 06825, USA; iversenm@sacredheart.edu

**Keywords:** microbiome, diet, lifestyle, physical activity, microbial diversity, microbial density

## Abstract

Recent discoveries of the purpose and potential of microbial interactions with humans have broad implications for our understanding of metabolism, immunity, the host–microbe genetic interactions. Bioavailability and bioaccessibility of phytonutrients in foods not only enrich microbial diversity in the lower human gastrointestinal tract (GIT) but also direct the functioning of the metagenome of the microbiota. Thus, healthy choices must include foods that contain nutrients that satisfy both the needs of humans and their microbes. Physical activity interventions at a moderate level of intensity have shown positive effects on metabolism and the microbiome, while intense training (>70% VO_2_max) reduces diversity in the short term. The microbiome of elite endurance athletes is a robust producer of short-chain fatty acids. A lifestyle lacking activity is associated with the development of chronic disease, and experimental conditions simulating weightlessness in humans demonstrate loss of muscle mass occurring in conjunction with a decline in gut short-chain fatty acid (SCFA) production and the microbes that produce them. This review summarizes evidence addressing the relationship between the intestinal microbiome, diet, and physical activity. Data from the studies reviewed suggest that food choices and physical fitness in developed countries promote a resource “curse” dilemma for the microbiome and our health.

## 1. Introduction

Microbial species in the gut, including fungi, viruses, and bacteria, are key influencers in human development, immune function, and health. The gut microbiome functions as both a nutritional competitor and supplier of nutraceuticals and facilitates human metabolism. In this way, the microbiome behaves as an economy that trades in nutrients and the end products of fermentation; thus, supply and demand forces may apply. Economists have observed that developing countries often suffer from the paradox of plenty, such that resource riches lead to marginalization and poverty in some groups [1]. The “resource curse” framework can be applied to the human microbiome, where richness in some nutritional components leads to bacterial overgrowth of certain species, thus crowding out the growth and sustainability of others [2]. The western pattern diet (WPD) due to its nutrient resources which are low in fiber and carbohydrate polymers of 10 or more monomeric units, disrupts microbial populations resulting in damage to the intestinal epithelial barrier (IEB)–the gatekeeper of nutrient permeability into the endothelium [3]. Once the intercellular junction has been breached, inflammation and tissue injury occur. Pathogens take advantage of this disruption, while commensal bacteria become displaced by dysregulation. Resource imbalance creates a lack of “leadership” among microbes and results in a lawless state of dysbiosis where some species go into decline.

The formation of the protective mucous barrier on the lining of the intestinal lumen creates a physical space for the host–microbe exchange of nutrients and metabolites [4]. In addition, the production of short-chain fatty acids (SCFAs) through the fermentation of polysaccharides supports the energy needs of endothelial cells [5]. Commensal bacteria also play a key role in the post-natal development of the gut-associated immune system by regulating immune homeostasis, thus providing another defense mechanism against invading pathogens [6]. The adoption of dietary patterns must be viewed within a broader context of a system of interactions between the host and its microbiome.

Physical activity (PA) has been shown to prevent and treat a number of chronic diseases, including heart disease, type 2 diabetes, high blood pressure, some cancers, depression, and dementia [7]. In an experiment using dry immersion to simulate extreme inactivity while controlling for diet, researchers observed that participants not only developed muscle loss, but experienced a reduction in microbial production of propionate and an increase in Firmicutes [8]. Studies on habitual athletes have shown that they also have high levels of Firmicutes, but individual species vary by sport and activity level [9]. The gut microbiome of sedentary adults in a short-term exercise intervention given supplemental protein developed an increase in microbial and viral diversity [10]. However, one would expect that when athletes are compared to sedentary individuals that bacterial species associated with health would be more abundant in the more active group, as well as a greater consumption of fruits and vegetables [9].

The literature on lifestyle patterns, including diet and physical activity, and their effect on the microbiome, identifies some differences by activity level and microbe–food matrix preferences and supports a resource environmental theoretical framework for future research. This scoping review seeks to review the role of diet and exercise-induced adaptations in shaping the gut microbiome to reveal the current state of our knowledge of their impact on health.

## 2. Methods

This scoping review of systematic is reported using guidelines from Joanna Briggs Institute Reviewer’s Manual and managed with JBI System for the Unified Management, Assessment, and Review of Information (SUMARI) [11,12].

### 2.1. Eligibility Criteria

As presented in the Table 1 PICOS chart, papers selected for inclusion were English language systematic reviews of cross-sectional, prospective cohort studies, randomized-controlled trials of either parallel or crossover design with healthy adult subjects aged 18 years or older. The exposure or intervention had to include diet pattern (Western-style, plant-based, vegan, omnivore, carbohydrate exclusion) and/or physical activity or exercise. Therapeutic interventions using pre- and pro-biotics as therapeutic agents were rejected, but studies using fermented whole foods were included. Studies should include comparisons between the type of diet and/or physical activity levels and changes in microbial composition of the gastrointestinal tract (GIT). Reviews examining probiotics outside of the context of dietary pattern and non-interventional reviews were excluded. Outcomes of the intervention or exposure should include gut microbiota composition through fecal samples and abundance composition or abundance of specific intestinal bacteria.

### 2.2. Search Strategy

A systematic search for systematic reviews was conducted on PubMed, CINAHL-EBSCO, Cochrane Database of Systematic Reviews, Prospective Register of Systematic Reviews (PROSPERO), PEDro, and PubMed was conducted in June 2022 from January 2012 to June 2022. The search aimed to identify systematic review papers investigating the influence of diet and/or physical activity on the microbiota composition of gut microbiota of healthy adults using a combination of MeSH and individual search terms. The search terms were microbiome OR microbiota OR gut microbiome in combination with diet AND/OR exercise AND/OR physical activity resulting in 282 articles after the elimination of duplicates. Study identification and selection are detailed in the PRISMA-ScR flow diagram (Figure 1) PRISM-ScR [13].

The initial searches produced several papers with questionable authority due to heterogeneity in reporting outcomes. Strengthening The Organization and Reporting of Microbiome Studies (STORMS) was developed from STROBE and STREGA (STrengthening the REporting of Genetic Association Studies) guidelines in 2021 to standardize the organization of essential information for papers reporting on the microbiome with the purpose of reducing reporting heterogeneity [14,15,16].

### 2.3. Study Selection and Data Extraction Process

Two reviewers (PGF and HF) independently screened the titles and abstracts, as well as assessed papers for eligibility. The Preferred Reporting Items for Systematic Reviews and Meta-Analyses (PRISMA-SCR) guideline was used as a basis for reporting. As study designs and outcome assessments varied, results are presented in a narrative way. Results are presented based on the PICOS criteria in Table 1 and PRISMA 2020 flow diagram in Figure 1 for new systematic reviews [13]. JBI Critical Appraisal Checklist was used to evaluate the risk of bias in the systemic reviews.

### 2.4. Results

Table 2 summarizes the characteristics of the included studies and the relevant results. Over time the quality of papers that reported on the effect of lifestyle on the gut microbiota has improved, but there remains an elevated level of heterogeneity in treatment outcomes making it difficult to generalize findings.

## 3. Microbiome

The microbiome is a collection of bacteria, fungi, and viruses that varies by location, age, health status, diet, and physical activity levels. The dominant bacterial phyla in the human microbiome are *Firmicutes*, *Bacteroidetes*, *Proteobacteria*, *Actinobacteria*, and *Verrucomicrobia* [44]. Most fungi are poor colonizers of the gastrointestinal tract (GIT), but seven taxa have been found: *C. albicans*, *Saccharomyces cerevisiae*, *A. niger*, *Penicillium* sp., *Pichia*, *Aspergillus*, and *Mucor* and are influenced by diet and alcohol consumption [45]. Candida is the most abundant genus in the first six weeks of life in infants, with colonization being greater in those delivered vaginally [46]. Bacteria reside in the gut, as well as other body sites (genital, skin, airway)—each with its own distinct population of archaea, viruses, and eukaryotes [47]. A healthy microbiome exhibits greater microbial diversity. Low microbial diversity is observed in disease states such as cardiovascular disease, cancer, obesity, and metabolic and immune disorders. The reduction in populations within the human microbial ecosystem or intestinal dysbiosis is associated with negative health effects [48].

The viruses (virobiota) and their communities (viromes) cross the IEB and are believed to have a cooperative evolution and symbiotic relationship with their host [49]. Viral colonization of the GI tract begins after birth and gradually increases with age [50]. Human endogenous retroviruses (HERVs) have entered host germ cells, eggs, and sperm, over successive generations of infecting our ancestors and make up approximately 8% of our genome [49]. HERVs can be coopted into protecting the host and participating in protective metabolic activities such as neuroprotection and embryonic development. Activation of HERVs can affect the expression of genes involved in immunity and inflammation. Researchers demonstrated that the fasting or overfed condition affected the expression of ERV-related genes in geese, suggesting that they are influential mediators in the development of non-alcoholic liver disease [51].

## 4. Diet

Individuals have unique gut microbial communities due to differences in host genetics, physical activity, aging, health, and dietary composition [52,53,54,55]. Food provides nutrients and other substrates for the bacteria that reside in the gut; in return, bacteria yields not only SCFAs, but significant amounts of vitamins K2 and B12, folate, riboflavin, thiamine, and other nutrients [56,57]. Interspecies bacterial competition for life-sustaining resources drives community composition [58]. To reduce dysbiosis and systemic inflammation, a better understanding of how strain diversity, their interactions, and the metabolism of various nutrients is crucial.

The habitual diet has wide-ranging effects on human health now and in the future. Telle-Hanson, Holven, and Ulven (2018) reviewed the literature on diet and its components, microbiota, and evidence of inflammation in dietary intervention trials to develop insight into the origin of cardiovascular disease [17]. Few conclusions were made due to the heterogeneity of factors and wide latitude in which the authors of the primary studies define a “healthy diet”. Data from the American Gut Project demonstrates that plant-based and flexitarian dietary patterns with higher Healthy Eating Index 2010 (HEI-2010) scores were associated with microbiome β-diversity or the extent of change in the number of microbial communities [59]. Diets rich in inflammatory foods such as refined grains, processed and red meats, fried foods, and added sugars are major factors in the causation of chronic diseases. Tools such as the Dietary Inflammatory Index (DII) are used to evaluate the inflammatory potential of an individual’s dietary pattern [60]. Fruits, vegetables, whole grains, and legumes consumption are all linked to the reduction of systemic inflammation as well as microbial diversity [61]. Consumption of whole fruits, vegetables, and legumes in a Mediterranean dietary pattern has been shown to increase fecal short-chain fatty acid (SCFA) levels, mainly due to fermentation of insoluble fibers by the most abundant phyla—*Firmicutes* and *Bacteroidetes* [62,63]. Grape and red wine, with their high polyphenol content, have been associated with beneficial changes in GIT microbial composition *Proteobacteria*, *Fusobacteria*, *Firmicutes*, *Bacteroidetes*, and *B. uniformis* [19]. The response to the consumption of 100% fruit juice varies by type, but orange juice has a positive effect on the gut microbiome [64,65]. The data from a short-term human trial using cherry juice suggests that participants who consumed a Western-style diet showed different gut microbiota responses than other participants due to their reduced ability to metabolize polyphenols [66]. Healthy volunteers in a month-long trial of Montmorency tart cherry concentrate had no change in species richness or microbial composition [67]. Thus, habitual dietary pattern affects the ability of individuals to receive all of the health benefits from fruits and vegetables.

In a study of food frequency and related bacterial genera in 98 participants, the authors reported that the habitual diet was correlated with separate clusters, termed “enterotypes,” primarily dominated by *Bacteroides*, *Prevotella*, and *Ruminococcus* [15]. The resulting dietary-enterotype framework showed that Bacteroides entero-type was highly associated with a more Westernized diet high in animal protein and saturated fats, while the *Prevotella* enterotype was associated with a carbohydrate-dominated diet. Vegetarians showed a mixed pattern of *Prevotella* and *Bacteroides*. Investigators in this controlled feeding trial demonstrated that by changing the diet pattern on a short-term basis to either a plant-based or animal-based diet, microbial community structure recovered to baseline bacterial composition once participants resumed their usual diet [68].

Nutrient-dense foods are higher in nutrients and lower in calories and naturally contain vitamins, minerals, fiber, or resistant starches. The MAL-ED study (Etiology, Risk Factors, and Interactions of Enteric Infections and Malnutrition and the Consequences for Child Health) followed a cohort of 1283 children ages 9 to 15 months from eight low resource populations to determine their risk of anemia, low retinol, zinc, ferritin, and high transferrin receptor (TfR) [69]. The researchers assessed diet intake, nutrient density, micronutrient status, and markers of inflammation. The authors concluded that after accounting for dietary nutrient density, there was an independent association between biomarkers of intestinal permeability and micronutrient status for children with anemia, low ferritin, and retinol levels. These data suggest that a nutrient-dense diet may be useful in reducing systemic inflammation.

A trial of a Mediterranean diet with an additional green tea supplement and minimal amounts of meat demonstrated improvement in cardiometabolic risk, weight loss, and changes in microbial abundance driven by a small low abundant non-core change in taxonomic composition [70]. The addition of a polyphenol-rich green tea supplement to the diet enhanced the abundance of *Prevotella* and *Bifidobacterium* involved in the synthesis and degradation of branched-chain amino acids. The presence of the genera *Prevotella* and *Bifidobacterium* is common in non-Western societies, and some authors have suggested that it may be a biomarker for a healthy lifestyle [71].

### 4.1. Western Diet

The Western-style diet, an energy-dense and nutrient-poor pattern, consists of processed meats, full-fat dairy such as cheese and ice cream, refined grains, and added sugars, which promote inflammation and chronic disease. A systematic review of 46 studies that measured the effect of Western food choices on inflammation concluded that “balanced” diets reduce the risk for chronic disease [72]. Across the 46 studies meeting inclusion criteria, individuals consuming Western-type and meat-based diets had higher markers of low-grade chronic inflammation. Results from studies on the MedDiet, a diet rich in fruits and vegetables, demonstrate that individuals following this type of diet have lower markers of low-grade inflammation. As these studies were drawn from cross-sectional observational designs, the authors concluded that in order to confirm these results, prospective trials are needed.

#### 4.1.1. Ultra-Processed Foods

One discriminating feature between traditional and Western dietary patterns is the consumption of ultra-processed foods. The Open Food Facts database, which gathers food product information from around the world, uses the NOVA classification system to score products [73]. This system of classification has been used in cross-sectional studies of diet intake and chronic disease [74]. The NOVA classification of food has four categories: unprocessed or minimally processed foods, processed culinary ingredients, processed foods, and ultra-processed food and drink products. A systematic review and meta-analysis of cross-sectional studies on the effect of exposure to ultra-processed foods (UPF) on health status using food frequency and 24 h recall data found an increased risk of overweight, hypertension, metabolic syndrome, and low high-density lipoprotein [75]. The authors also reviewed five quality prospective-cohort studies and found an increased risk of all-cause mortality (RR 1.25, 95% CI 1.14, 1.37; *p* < 0.00001) for individuals who consumed elevated levels of UPF.

Food processing adds sodium, sugar, and other additives to modify the flavor, texture, and/or color. The influence of high sodium diets (HSD) is attributed to aberrant T-cell activation resulting in essential hypertension and other autoimmune diseases [76]. Experimental evidence using a rat model suggests that dietary sodium also alters the composition of gut microbial taxa, specifically *Christensenellaceae* and families, as well as the *Erwinia* and the *Anaerostipes* genera [77]. HSD also elevated levels of proteinuria and produced hypertension, a finding in agreement with human studies [78]. A randomized, double-blind, placebo-controlled crossover trial of a reduced-sodium diet in 145 untreated hypertensive individuals ages 30 to 75 years demonstrated increased circulating levels of SCFAs and decreased blood pressure [79]. Results for women showed a significant difference in all eight SCFAs, but there was no difference by race. In another study, a high sodium diet consumed by healthy volunteers was associated with an increase in blood pressure and an increased abundance of *Prevotella*, *Bacteroides*, and *Ruminococcaceae* [80]. Since “enterotypes” associated with diet have been described as being either *Prevotella* or *Bacteroides* dominant, this may suggest that there is a transition point from healthy to unhealthy, making this shift a marker for wellness [71].

Components of a processed diet may include the use of additives, particularly titanium dioxide (TiO_2_) and emulsifiers. These components influence the integrity of the IEB, while food additives promote biofilm formation, altering microbial composition [81,82,83,84,85,86]. Another role of food processing is the enrichment of foods with vitamins, minerals, and nutraceuticals according to regulations or for product marketing. By increasing nutrient density, fortification promotes biofilm formation under these conditions and promotes bacterial persistence to antibiotics.

Consumption of ultra-processed foods may induce low-grade gut inflammation and increase an individual’s risk for cancer and cardiovascular disease. An online survey of consumers found that they readily identified ultra-processed foods as foods being ready-to-eat and containing additives, food colorings, preservatives, and stabilizers [87]. Low-income consumers choose foods that are ultra-processed for several reasons, such as cost, convenience, and taste [75,88]. Public health educators should focus on educating consumers about how to make smarter choices by reading the food label and choosing products with few additives and minimal processing.

#### 4.1.2. Protein

The degradation of dietary protein occurs through human and microbial proteases. Bacteria rely on carbohydrates and fiber for energy, but they are flexible and will use protein as an energy source when carbohydrate resources are low. Proteolytic fermentation produces phenols, branch-chain amino acids, sulfides, indoles, and ammonia as by-products [89]. Unlike the products of carbohydrate fermentation, colonic protein metabolites are often associated with colon cancer [90]. Researchers in one controlled human trial found that a three-week high protein diet had the effect of regulating the production of bacterial metabolites, causing an increase in the degradation of amino acids in the gut and modifying gut mucosal genes involved in cell cycle regulation [91]. Using a Human Intestinal Microbial Ecosystem (SHIME(R)) model researchers demonstrated that a high-protein diet (2.5 g/L casein) produces a different microbial community composition than a high-fiber (0.6 g/L casein) diet [92]. High-protein diets fed to mice lead to the development of obesity which was a major factor in shifting the gut microbial populations, but this effect was dependent on the diet’s protein-to-sucrose ratio [93].

One outcome of consuming diets rich in the essential amino acid tryptophan, such as dairy, poultry, and nuts, is that they favor the generation of indole and indole derivatives by colonic bacteria. Indole is a signaling molecule that regulates the host immune system by supporting epithelial barrier defense in a way that controls pathogens without producing an inflammatory response [94,95], thereby reducing chronic inflammation caused by the passage of bacteria and toxins from the gut into the bloodstream [96].

#### 4.1.3. Fats

Trans-fat, or trans-fatty acids, are a by-product of the hydrogenation of vegetable oils to enhance shelf-life and other desirable food characteristics. Once considered to be a healthier alternative to saturated fat, there is now a consensus on the role trans fatty acids play in the development of heart disease and stroke by increasing LDL cholesterol and reducing HDL cholesterol. The association of high intakes of iTFA *(industrial trans-fats)* with cardiovascular disease has led to public health recommendations to reduce or eliminate their use. High intake of iTFA in mice has been shown to cause significant dysbiosis of gut microbiota [97]. A survey of the consumption of industrial trans-fatty acids reported that 22 out of 29 countries sampled found the intake of total trans-fat is currently below recommendations of 1% [98].

The ability of colonic bacteria to adapt to a fats-only source of energy for bacterial growth has been demonstrated in a fats-only medium through genes encoding for enzymes involved in fat degradation [99]. In the fats-only condition, *Alistipes* spp., *Bilophila* spp., and total *Proteobacteria* were favored, while SCFA-producing species, such as *Bacteroides*, *Clostridium*, and *Eubacterium* spp., declined. Through the use of an in vitro human gut simulator (HGS) model, the authors also demonstrated that in a fats-only medium fortified with micronutrients that lack protein or carbohydrate, bacterial communities responded with a change in community structure [99]. Colonic bacteria adaptation to a fats-only diet comes at a price to the host through a reduction in the absorption of antioxidants and an increase in the expression of bacterial virulence factors [100]. In a systematic review of the MyNewGut project data, researchers found that diets high in saturated fats degreased both richness and diversity, while those high in monounsaturated fatty acids (MUFA) produced a lower total number of bacteria, whereas a diet rich in polyunsaturated fatty acids (PUFA) resulted in no change [21].

#### 4.1.4. Carbohydrates and Fiber

WPDs are comprised of highly refined sugars, particularly sucrose and fructose. It has been suggested that one of the causes of obesity may be an adaptation in microbial diversity due to one or more components in the Western diet [101]. Researchers using a mouse model reported that a diet high in fructose and/or sucrose, a pellet with 55% fructose-42% glucose ratio, alters the ratio of *Bacteroidetes* to Proteobacteria, favoring the profile associated with metabolic syndrome [102]. Metabolic syndrome increases an individual’s risk for cardiovascular disease and type 2 diabetes. Lifestyle intervention research using diet and physical activity approaches has only been partially successful in controlling the epidemic of obesity.

Diets high in fermentable carbohydrates and fiber alter the nutritional ecology of bacteria resulting in bacterial diversity, particular species that produce SCFAs such as butyrate, acetate, and propionate, as well as lactate and gasses (CO_2_, H_2,_ and CH_4_) [89]. In a study of pregnant women, low fiber intake was associated with a gut microbiota profile favoring the fermentation of lactate and insulin resistance, while a diet rich in fruits and vegetables promoted bacteria that produce SCFAs through bacterial fermentation of plant polysaccharides [103]. Incorporating healthier eating behaviors into prenatal counseling may support a more beneficial gut microbiome.

Naturally sweet foods such as honey have long been known to have both antimicrobial and probiotic properties, although it is primarily glucose and fructose [104]. Honey promotes commensal strains such as *Lactobacillus reuteri*, *Lactobacillus rhamnosus*, and *Bifidobacterium lactis* while limiting the growth and adhesion of pathogenic bacteria in the host. North American maple syrup is rich in phytochemicals and oligosaccharides, as well as lignin [105,106]. Maple syrup improved insulin sensitivity and reduced non-alcoholic fatty liver in a diet-induced obese insulin-resistant rat model [107]. A randomized controlled trial in humans on the role of maple syrup on gut microbial diversity and metabolic syndrome in humans is currently underway [108].

Fermented foods are processed using techniques for “controlled microbial growth” to produce enzymes that alter food characteristics [109]. Food and beverage products have traditionally been either fermented, aged, or inoculated with bacteria and yeasts to preserve and enhance their flavor. Studies on the impact of fermented foods on the gut microbiome suggest they have a positive impact on the gut microbiome, but the quality of the evidence is unclear [110]. The generalizability of health benefits from fermentation is limited due to the undefined microbial content of starter cultures and the uniqueness of each processing plant’s bacteria and fungi communities [111]. The fermentation process has potential positive impacts, such as the release of bioactive peptides, biogenic amines, and phenolic compounds with increased antioxidant activity. Bioactive peptides and polyamines have beneficial effects on cardiovascular, immune and metabolic health, such as enhanced mineral absorption and reduced oxidative stress [112].

In addition to acting as the food source for intestinal bacteria-resistant starches, polyphenols and fiber have been shown experimentally to modulate the quantity and composition of microbiota by inducing prophages, virus-like-particles (VLPs) [113]. These bacteriophages protect the gut lining from pathogens and insert into the genome of their host’s chromosomal DNA as prophages. The bactericidal effects of VLPs are triggered by several common foods and products such as Tabasco sauce, vinegar, Kombucha, cinnamon, miso, oregano, coffee Arabica, and stevia. Although stevia appears to have positive effects on gut microbiota, it can induce heritable changes in gene expression in offspring. Experimental evidence of a recent study on the effect of maternal stevia consumption on obese rats fed a diet high in fat/sugar (HFS) combined with either aspartame or stevia resulted in second-generational effects of obesity and glucose intolerance in the offspring [114]. The proposed mechanism for the observed effect was the development of altered gene expression in the mesolimbic reward pathway associated with feeding behavior.

In the past, the mechanism behind the observed beneficial health effects of cultural superfoods, such as natto and kimchi, was unknown [115]. In the future, diets may be personalized to enhance the ecology of the gut microbiome through the targeted use of these foods.

#### 4.1.5. Diet Supplementation

The quality of the diet shapes the health of the host and gut. For example, vitamin A deficiency has an effect on microbial community structure and gene expression [116]. The development of the gut microbiome in infancy and childhood is affected by the supply of micronutrients in the diet [117]. In regard to micronutrient supplementation, a study of Kenyan infants concluded that iron fortification adversely affects the gut microbiome by significantly increasing the population of pathogens, such as *Clostridium* and *Escherichia/Shigella*, with a reduction in beneficial *Bifidobacterium* which increases inflammation [118]. The authors of the study concluded that when iron is poorly absorbed by the small intestine, the resulting iron overload in the colon disrupts the bacterial population. The effect of iron and zinc fortified foods on gut microbiota was reviewed in one systematic review, and the authors reported no adverse effects in the five studies retrieved [119]. With the widespread fortification of food products to promote health and prevent nutrient deficiencies, more research on the effects of biofortification on gut bacterial populations is needed.

#### 4.1.6. Water

Data from the American Gut project database suggests that drinking water source is associated with microbiota composition [120]. Fecal samples from individuals who consumed well water had greater β-diversity compared to those who drank bottled, filtered, or tap water. Dai et al. reported that microbes coming from drinking water supplies that have been disinfected versus non-disinfected water have a less diverse structure and function, favoring microbes that utilize fatty acids derived from microbial decomposition [121].

#### 4.1.7. Plant-Based Diets: Vegetarian and Vegan

Vegetarian and vegan diets rich in cereals, nuts, fruit, vegetables, and legumes are the most studied in regard to gut microbial diversity and health effects. The long-term dietary pattern an individual chooses to follow will have an effect on the gut microbiota. Each dietary pattern has a different profile of gut microbiota producing distinct types of postbiotics, the compounds produced by bacteria residing in the gut that support health.

Consumption of a diet rich in plant-based foods rich in fiber, folate, and carotenoids improves resistance to inflammation and oxidative stress, promoting healthy cellular aging [122,123]. Data from the Nurses’ Health Study suggests that a diet rich in dietary fiber from cereals is positively associated with leukocyte telomere length [124]. The literature on nut consumption and the microbiome is inconsistent, with one review concluding no effect on β-diversity and increases in *Clostridium*, *Lachnospira*, and *Rosburia*, producers of SCFAs [27]. The *Firmicutes/Bacteroidetes* ratio is lower in vegans compared to omnivores and differs according to their metabolic profiles [33,125]. Meanwhile, another reviewed almonds, walnuts, hazelnuts, or pistachios, limiting the included papers to those with next-generation sequencing technology [29]. Almonds, hazelnuts, and pistachios had a small impact on microbial diversity, but walnuts were superior in changing both α- and β-diversity. The pulses are the edible seeds of the legume plant and include beans, lentils, and peas. A limited number of studies have looked at the effect of pulses on the microbiome, but those that do exist have found that whole pulses and pulse-derived flour have positive effects on diversity and richness [28]. Little is known about the components of each type of pulse and their prebiotics, fructo-oligosaccharides (FOS) and galacto-oligosaccharides (GOS) [126].

Adult whole milk consumption has been associated with an increased risk of chronic disease and reduced telomere length. Using NHANES data, a total of 3072 women and 2762 men were included in an analysis of milk consumption and telomere length [127]. Telomere length did not differ by how frequently men and women drank milk, but adults who drank full-fat or 2% milk had significantly shorter telomeres than those consuming nonfat or 1% milk, and that difference was clinically meaningful. This finding is consistent with the dietary guidelines, which encourage low-fat dairy and discourages the consumption of full-fat milk for adults. The effect of unfermented dairy on gut microbiota was a reduction in richness and diversity, while yogurt and kefir increased both *Lactobacillus* and *Bifidobacterium* [26]. Habitual vegans and vegetarians, when compared to omnivores, have a greater abundance of fiber-degrading species and lactic acid bacteria. Yet, when vegan and vegetarian diets were compared, no consistent difference in microbiota composition was found [18,32]. Both diets are rich in polyphenols that increase *Bifidobacterium* and *Lactobacillus*.

Diets excluding grains, dairy products, salt, and refined sugar, also known as ketogenic or modern paleolithic diets, have become popular in the past few years. A study comparing the gut microbiome profiles of 15 urban-dwelling Italians practicing a paleolithic diet to populations following a Mediterranean diet and traditional hunter-gatherer groups found major differences between groups, including a relative abundance of *Bacteroides* and other organisms in the paleo diet sample typically associated with high animal protein and saturated fat diets and an altered bile acid profile associated with inflammatory bowel and colon cancer [128]. In a review of RCTs, studies using whole-grain versus low-fiber diets found that fiber increases microbial diversity, abundance, and beneficial metabolites [20,23]. Gluten-free diets in healthy subjects accompanied by a reduction in resistant carbohydrate intake results in lower levels of *Bifidobacterium*, *B. longum*, and *Lactobacillus* and more unhealthy species [129,130]. Based on this information, dietary adherence to paleolithic diets over the long term would not be advised.

#### 4.1.8. Mediterranean

The CARDIVEG (Cardiovascular Prevention with Vegetarian Diet) study, a three-month crossover design comparing the Mediterranean to a vegetarian diet, found that there was no statistical difference in microbial diversity and no change in the *Firmicutes/Bacteroidetes* ratio between the two diets [58]. Short-term dietary changes produced only small variations in gut microbiota composition. In another study, habitual Mediterranean diet and vegetarian diet consumption were associated with increased levels of fecal SCFAs, *Prevotella* and *Firmicutes* [10]. The effects of the Mediterranean diet on the gut microbiome, in general, have a positive impact on the *Firmicutes/Bacteroides* ratio, but results were not consistent across studies in a recent systematic review of prospective and cross-sectional studies [37]. The authors suggest that this may be due to differences in diet adherence between participants. The PREDIMED-Plus study looked at the influence of an energy-restricted Mediterranean diet along with physical activity on the microbiome of older adults [131]. Controls were given instruction on the Mediterranean diet with no calorie restrictions or physical activity guidance. First-year results reported that the physically active and calorie-restricted group had a significant increase in the *Bacteriodetes/Firmicutes* ratio and greater weight loss than controls. However, both groups saw a shift to more SCFA-producing genera, suggesting that the Mediterranean diet selectively enhances specific anaerobic gut bacteria responsible for the fermentation of complex carbohydrates.

Seafood is an important part of the Mediterranean, contributing to the intake of omega-3 fatty acids [132] but also influences the microbiome. In a crossover design study comparing the metabolic markers and fecal bacteria before and after a diet with and without lean seafood, lean seafood resulted in higher serum TMA and fecal TMA, presumably due to increased bacterial degradation of TMAO [133,134]. The fecal microbiota analysis showed an increase in *Firmicutes* and decreased *Bacteroides*, as well as higher amounts of *Clostridium Clusters IV*, producers of butyrate in the gut. Four weeks after the lean seafood intervention, there were significant improvements in cardiovascular risk factors, HDL, and a decrease in circulating triacylglycerol (TAG).

## 5. Physical Activity

In general, physical activity both lowers *Bacteroidetes* and raises *Firmicutes*. Moderate physical activity, defined as endurance activity performed 3 days a week for 30 to 60 min at 60% of heart rate reserve, increases both microbial diversity and richness, but authors concluded in one systematic review that the overall strength of the evidence is weak due to poor control over dietary patterns confounding the data interpretation [22]. A review of observational studies showed that only 5/9 showed that a higher level of physical activity or cardiovascular fitness was associated with greater α-diversity [24]. The effect of exercise, in general, produces an increase in *Bifidobacterium*, *Lactobacilli*, and *Akkermansia* [135]. Athletes have higher microbial diversity and produce an abundance of short-chain fatty acids (SCFA) [22]. Fecal and urine samples from elite cyclists participating in the 2016 Olympics were characterized by different training loads and amounts of time spent in static and dynamic activities [136]. Diet intake did not vary significantly between sports classification groups. The analysis demonstrated differences in microbial composition and SCFA metabolites detected in both feces and urine. A study of recreational and professional cyclists reported an expected abundance of Bacteroides, *Faecalibacterium*, and *Eubacterium*, along with increased levels of *Methanobrevibacter smithii*, indicating an upregulation of energy pathways using methane, a high-energy fuel [137]. This adaptation in athletes is beneficial because methanogens, such as *Methanobrevibacter smithii*, must produce methane to conserve energy for growth rather than produce it as a by-product of metabolism [138]. Mohr and colleagues have published a comprehensive review of the literature on the gut microbiota of various sports [135].

Estaki et al. examined the fitness levels of young adults as measured by VO_2_ peak and fecal microbiota and found that an individual’s fitness category was responsible for 20% of the variation associated with taxonomic richness [139]. The authors examined diet as a confounding variable and found that only one component was significant for beta diversity—protein. Protein intake and age accounted for an additional 7.9 and 2.3%, respectively, of the community beta diversity. A systematic review of 10 studies examining the influence of physical activity on the human gut microbiota composition concluded that most studies did not account for the influence of diet, and further studies of dietary protein levels are indicated to confirm this finding [31]. During periods of intense activity, the gut wall junctions lose their integrity, permitting materials from the lumen to translocate into the bloodstream resulting in increased inflammation due to IL-6 production [53]. Overall, the shift in the microbiome varies by the conditioning level of the athlete, with well-conditioned individuals benefiting from species involved in metabolic pathways producing amino acids and the metabolism of carbohydrates and fiber [53]. The reciprocal relationship between diet, activity, and microbial communities, which has been labeled the “nutrition-microbiota-physical activity triad”, represents a regulatory role for microbial communities [140].

Moderate to vigorous activity of less than an hour enhances anti-pathogen activity through the release of NK cells and CD8+ T lymphocytes, which over time improves the monitoring ability of the immune system to detect pathogens and tumors [141]. When conditioned athletes were compared to others who met the general recommendations for exercise, the less conditioned group showed no meaningful change in microbial richness or diversity [34,36]. A review of both interventional and observational studies found no consistency between trials for changes in microbial diversity, but there were decreases in pathogenic species and increases in beneficial taxa [9,35]. These studies support the conclusion that the benefits of exercise to the microbiome and its influence on host immune function are on a continuum, but the optimal has yet to be determined [142].

## 6. Discussion

### 6.1. Introduction

The aim of this scoping review was to identify lifestyle factors influencing the gut microbiome and host health. The microbiome is a dynamic system that affects not only the host’s internal environment but is also driven by external factors such as the genome and the health behaviors of the human host [143]. We have argued that resource availability is a major driver of microbial community composition and sustainability.

However, no one model can capture or predict all of the host–microbiome-lifestyle interactions. Stegan, Bottos, and Jansson have put forth a unified conceptual framework upon which to study time-dependent interactions, which may lead to manipulating microbes to support health and human performance [144]. Within this framework, an individual’s history of diet and physical activity influences the internal dynamics of the community structure and dynamics. External forces, such as resource availability, impact community density, and composition. The use of data-intensive modeling using the triad of history, external forces, and internal dynamics will expand our understanding of microbial communities and human health.

### 6.2. Limitations and Strengths of This Review

The authors were challenged by the pace of change in the emerging fields of microbiology within nutrition and exercise science but endeavored to find the highest level of evidence to begin integrating this knowledge for public health researchers and educators. The strength of this review is its recency, and its weakness is that the heterogeneity in the technologies and reported outcomes by the authors make data extraction for a meta-analysis impossible at this time. The difficulty in making general recommendations is that it was rare to find a physical activity paper within our search criteria that controlled for diet [31,131]. Animal studies demonstrate that exercise performance and conditioning that takes place in long-term interventions have positive effects on both the microbiome and host metabolism [145]. Transferring this model to human studies will prove challenging because of the need to maintain a controlled research diet over the length of the study. The “captive” environment needed to perform research in animals may reduce the diversity of the microbiome by limiting exposure to other mammals and the environment [146]. Thus, systematic reviews of studies using natural experimental designs on well-defined human sub-populations may yield the best available evidence.

## 7. Conclusions

Promoting a healthy gut microbiome is crucial to protecting and supporting the health of an individual. Food and activity choices can have a dramatic impact on the gut microbiome. In turn, lactate acid bacteria may have a positive influence on sports performance and recovery through the conversion of lactate to propionate. Diets that are based on the Healthy Eating Index, such as the Mediterranean diet, promote the development of gut microbial diversity, which provides beneficial postbiotics such as SCFAs and butyrate. Polyphenols, found in abundance in plant foods, yield microbial metabolites that reduce inflammation and promote human health. Including prebiotics and probiotics as a functional food-based approach to manipulating the microbiome may prevent or reduce the progression of diet-related diseases. When making food choices, individuals should follow a diet that is rich in fruits, vegetables, whole grains, legumes, nuts, and seeds and low in processed foods, refined grains, and meats. Future versions of the Physical Activity and Dietary Guidelines for Americans should also review the literature on the impact of current recommendations for exercise on the abundance and diversity of the microbiome [147,148]. Our current state of knowledge on how changes in the microbiota determine long-term health and the aging process is lacking. Researchers manipulating the microbiome with a targeted probiotic approach should consider how microbial dynamics interplay with diet and physical activity in the study design.

## Figures and Tables

**Figure 1 ijerph-19-08361-f001:**
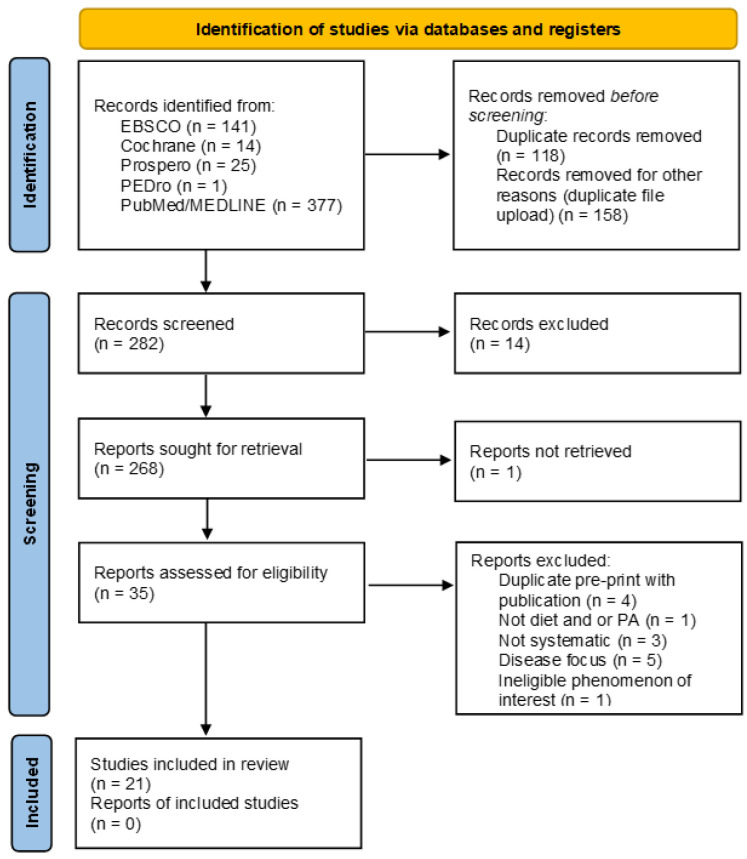
PRISMA-ScR flow diagram.

**Table 1 ijerph-19-08361-t001:** PICOS criteria for inclusion.

PICOS Format	Description
Population	Healthy subjects Adult humans aged 18 years or older
Intervention or exposure	Diet (Western-style, plant-based, vegan); and/or Physical activity or exercise
Comparisons	Diet (omnivore, Western-type, vegetarians, vegans) and/or Physical Activity level; reviews of interventions examining probiotics solely and non-interventional reviews were excluded
Outcome	Gut microbiota composition through fecal samples; abundance composition or abundance of specific intestinal bacteria
Study design	Systematic reviews of cross-sectional, prospective cohort studies, randomized-controlled trials of either parallel or crossover design; and reviews or studies for background information on food or physical activity

**Table 2 ijerph-19-08361-t002:** Summary of systematic reviews and metanalyses on the effect of diet or exercise on the microbiome.

Aim and Design of Studies	Number of Studies	Quality	Effect on Microbiome	First Author/Study Name
Influence of a healthy diet pattern on the microbiome and inflammatory markers Interventional human trials	18	Critical appraisal not reported	Due to heterogeneity in study design and type of subjects, no conclusions could be made	Telle-Hansen (2018) [17]
Vegan and vegetarian diet association with gut microbiota composition Cross-sectional/cohort/RCT	37	^a^ Newcastle–Ottawa scale NOS: 4.6 out of 10 points	No consistent association between a vegan or vegetarian diet and microbiota composition	Trefflich (2019) [18]
Effect of wine and grape polyphenols on human gut microbiota. Meta-analysis RCTs	7	^b^ Cochrane Risk of Bias Low (5) (1) unclear (1) high risk	Increased *Proteobacteria*, *Fusobacteria*, *Firmicutes*, *Bacteroidetes*, and *B. uniformis* after red wine intake; Decrease in dysbiosis-associated species: *Clostridum*, *Eubacterium*, and *Bacteroides*	Nash (2018) [19]
Dietary fiber intervention on microbiome Meta-analysis of RCTs	64-reviewed 58-retained for meta-analysis 12 studies focused on whole-grain diet versus a low-fiber diet	^b^ Cochrane Risk of Bias Low to moderate risk of bias (*n* = 64)	Dietary fiber intervention compared to placebo/low fiber diet did not significantly increase α-diversity, but increased abundance of *Bifidobacterium* spp. No difference in *Lactobacillus* spp. Abundance with food intervention, but significant in fiber supplement group	So (2018) [20]
Dietary fat and gut microbiota Cross-sectional, cohort; interventional studies and randomized controlled trials	16	^b^ Cochrane Risk of Bias 14 RCT-low risk; 2 RCT-high-risk ^a^ Newcastle-Ottawa NOS: 3 = 8, 3 = 7, 1 + 6, and 2 = 5	n3, n6 PUFA increase beneficial bacteria; high fat/saturated fat diets reduced richness and diversity and had negative metabolic health outcomes; observational studies show an association between fat and health outcomes	Wolters (2019) [21]
Association between exercise and gut microbial composition in mammals RCT, cross-sectional, and cohort studies	Human—20 Animal—5	^b^ Cochrane Risk of Bias–unclear Low Quality Lack of appraisal tools for heterogeneous models	Exercise was associated with changes in gut microbial composition, an increase in butyrate-producing bacteria, and fecal butyrate	Mitchell (2019) [22]
Effects of intact cereal grain fibers on microbiome RCT, RCT crossover, non-randomized	40	Critical appraisal not provided	Cereal fiber (6–8 g) increases diversity and abundance; increase in bacterial metabolites	Jefferson (2019) [23]
Influence of exercise on the human gut microbiota in healthy adults Observational and case-control	18	^d^ PEDro 18—Medium	4/9 observational studies showed higher levels of physical activity or cardiorespiratory fitness were positively associated with α-diversity	Ortez-Alvarez (2020) [24]
Influence of endurance training intervention and gut microbiome Interventional studies > 4 weeks duration	5	^d^ PEDro 4 studies score ≥ 4—Fair quality 1 study was rated Poor	PA significantly lowers abundance of Bacteroidetes and increases Firmicutes and β diversity in some studies	Shahar (2020) [25]
Effects of dairy and dairy derivatives on the gut microbiota (Bovine, yogurt, soy)	8	^b^ Cochrane risk-of-bias 2—low 5—some concerns 1—high risk	Richness and diversity declined in all types of milk, *Lactobacillus* increased in bovine milk; fermented yogurt and kefir increased *Lactobacillus* and *Bifidobacterium*	Aslam (2020) [26]
Effect of nut consumption on gut microbiome and gut function RCTs	8	^b^ Cochrane risk-of-bias No studies were low risk of bias; variable across categories of analysis	Meta-analysis found no effect on β-diversity; no effect of nut type, dose, duration of intervention; increased abundances of *Clostridium*, *Lachnospira* and *Roseburia*	Creedon (2020) [27]
Effect of dietary pulses on microbial populations RCT-C, (cross-over); Interventions with control or placebo group	5	Critical appraisal not provided	Bacteroides fragilis OUT↓ ^i^ for navy bean pulse flour; No difference in Shannon index for diet with chickpeas; lupin fiber consumption decreased abundance of Bacteroides-Prevotella	Marinangeli (2020) [28]
Effect of nut consumption on gut microbiome RCT-C, (cross-over); RCT parallel design, and pre/post-test studies	8	^f^ Quality Criteria Checklist and ^g^ Risk of Bias Assessment Tool 6/8 positive quality 2 neutral	Nuts in general, but especially walnuts, had an impact on gut microbial composition	Fitzgerald (2021) [29]
Association between physical activity and changes in intestinal microbiota composition Cross-sectional and longitudinal studies	17	^h^ ROBINS-I 6—low 7—moderate 1—serious 3—not reported	Increase in SCFAs concentration after the training period in lean athletes only; composition and diversity differ by sport	Aya (2021) [30]
Physical activity influences on human gut microbiota independent of diet observational	10 4/10 studies controlled for diet	^e^ JBI Critical Appraisal Checklist-criteria met ^b^ Cochrane Risk of Bias −2/20 some concerns	Variability is affected by dietary factors and physical characteristics; use of high protein diets contributes to greater variability among athletes	Dorelli (2021) [31]
Dietary habits and gut microbiota in healthy adults cross-sectional and RCT Diet regimen studies	16	^a^ Newcastle-Ottawa scale Mean score for cross-sectional studies was 5/10; ^b^ Cochrane Collaboration tool risk of bias-Low	Significant impact on some bacterial genera from a rich and varied omnivore diet, such as Mediterranean	Gibiino (2021) [32]
Vegan diet and gut microbiota Cross-sectional studies	9	^a^ Newcastle-Ottawa scale Most studies scored “medium” quality	*Firmicutes/Bacteroidetes* ratio is lower in vegans compared to omnivores; Abundance of *Bacteriodetes* and *Prevotella* in vegans	Losno (2021) [33]
a- and β-diversity in obese and non-obese adults Intervention studies and RCT	32 22 reported Shannon Index (diversity) 25 studies investigated diversity; 2 did not; 5 did not stratify by BMI	^h^ Adapted ROBINS-I Serious risk in one domain: 22 Moderate: 10	Higher levels of PA and cardiorespiratory fitness are associated with greater α-diversity and increases in some phyla and certain short-chain fatty acids	Pinart (2021) [34]
Effects of exercise and physical activity on the gut microbiome in older adults Observational and interventional studies	7	Critical appraisal not provided	PA had beneficial impact on the gut microbial composition of older adults	Ramos (2022) [35]
Physical activity and human gut microbiota in healthy and unhealthy subjects Observational and interventional studies	25	^h^ ROBINS-I 8 studies scored 4–5 ^c^ Jadad Scale 4/5 studies: Moderate ^e^ JBI Critical Appraisal Checklist for Analytical Cross-Sectional Studies 12/12 Included	No significant change in richness and diversity in gut microbiota for minimum PA recommendations Microbial diversity is associated with aerobic exercise	Cataldi (2022) [36]
Effect of MedDiet on microbiota and metabolites RCT and Observational studies	34 17-RCT 17-Observ	^b^ Cochrane (RCT) Mixed Quality ^a^ Newcastle-Ottawa scale (Observational) Prospective Studies: High 2 Moderate 1 Low 2 Cross-sectional High 6 Moderate 6 Low 2	Overall positive impact of Mediterranean diet on Firmicutes/Bacteroidetes ratio. but effects are not consistent between studies due to adherence differences and fewer species that utilize oligosaccharides and simple sugars	Kimble (2022) [37]

^a^ Newcastle–Ottawa scale (NOS) [38]; ^b^ Cochrane risk of bias [39]; ^c^ Jadad Scale or Oxford Quality Scoring System Scale [40]; ^d^ PEDro scale (Maher, 2003); ^e^ JBI Critical Appraisal Checklist for Analytical Cross-Sectional Studies [41]; ^f^ Quality Criteria Checklist; ^g^ Risk of Bias Assessment Tool [42], ^h^ ROBINS-I [43] and ^i^
^↓^ indicates a decrease.

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
