# Peer review of "Essential Factors for a Healthy Microbiome: A Scoping Review"

_ijerph, 2022, doi:10.3390/ijerph19148361_

Round 1

Reviewer 1 Report

Although the review is rich in quality references and effectively related to the text, the absence of a methodologic description of performing the bibliographic search, inclusion criteria and data analysis is critical. As stated by the journal, reviews should follow the Prisma Statement Guidelines. Any review presented to IJERPH should bring an advance and a specific contribution to the scientific research field.

It should be considered a reasonable explanation of the search procedure, databases consulted, etc. The references that are used to support the evidence should follow specific criteria for their inclusion. Although in a qualitative synthesis, the data analysis should be performed in a way that answers the review goal. In this case, the review only presents an overview of the common knowledge, without a specific goal or the required systematization of the results that create evidence.

The structure of the document is also imbalanced. The results are not clearly presented, and the discussion summarises the main findings for future work, not providing an appropriate integrative analysis of the results.

Author Response

Dear reviewer,

I have completed a systematic scoping review according guidelines from Joanna Briggs Institute Reviewer’s Manual and managed with JBI System for the Unified Management, Assessment, and Review of Information (SUMARI). I am in the process of mapping these results into the revision and getting responses from my co-authors. The methodology is completely detailed with Appendices if desired. I will revise the discussion as you suggested.

I have not met my deadline, but am asking for additional time. 

Reviewer 2 Report

Even though the idea sounds really interesting, some essential points need more information, clarification and rewriting to improve this article.

Major points

In accordance with the topics covered in this manuscript, the title should improve. It would be better to “Essential factors for a healthy microbiome: a review

The main objective must be the same throughout the manuscript in the abstract and introduction. In the abstract: Only the first time an abbreviation appears, the full name must be entered. Authors should not use the words that appear in the title as keywords. The references are recent and relevant.

ü  The authors said “Nutritional deficiencies are a result of mismanagement or inability to adapt to changing conditions” in Western countries?

ü  How PA is a fundamental tool in the prevention and treatment of several chronic diseases?

ü  Why is Table 1 important in the topic? The authors did not mention it in the text.

ü  What do the microbes represent (Lines 187-188).

ü  …antibiotic resistance? (Lines 195-196).

ü  …bacterial genes? (Line 214).

ü  …amino acid tryptophan…for example?

ü  What is the expected ratio (Lines 253-254).

ü  The point “Micronutrients” is weak, the authors could provide more information on other micronutrients.

ü  What do you mean by β-diversity?

ü  It is unclear what the authors intended to convey with Figure 1.

ü  Prebiotics and probiotics, why they have not been mentioned before in the text.

The discussion should be more argumentative.

The conclusion needs improvement and should be the same as the summary.

Author Response

Reviewer #1 asked me to add a section on methodology. This caused a delay as I ended up doing an updated (2012 to June 2022) scoping review of systematic reviews according to using guidelines from Joanna Briggs Institute Reviewer’s Manual and managed with JBI System for the Unified Management, Assessment, and Review of Information (SUMARI). Now we have a flow chart and are in the process of mapping the results into the paper. Thus, we are asking for more time. 

Reviewer 3 Report

Major limitations:

1. The insertion of the mediterranean diet inside the "plant-based" chapter and the unification with vegetarian and vegan diets is not correct. Authors should separate the observations on mediterranean diet separately from vegetarian and, most importantly, vegan diet for a better understanding of each entity

2. In chapter 4.1.2 authors refer to a "high protein" diet, given that there is no consensus on how much protein this is, they should state the gr/kg or percentage for a better understanding. Also, references to fat content should be reserved for the specific paragraph

3. The sugar content of fruit may have an impact and varies if the fruit is consumed as whole or in the form of juice. Authors should better clarify if the studies specified this difference

4. In the abstract and introduction, authors refer to the impact of PA on microbiome, but in the text there is no specific paragraph and the mentions are really concise. Authors should expand this section or state clearly the limitations in the text

4. In the conclusion, authors advocate for a plant-based diet but the data presented suggests similar (if not better) results with a mediterranean diet so they should correct this statement

5. Text needs some grammar fix: some sentences (i.e. line 54-57) have a repetition of the subject and verb both at the beginning and at the end

Author Response

Reviewer #1 has asked for a methodology section and other revisions that we are working on. We decided to update with a scoping review of systematic reviews using the guidelines using guidelines from Joanna Briggs Institute Reviewer’s Manual and managed with JBI System for the Unified Management, Assessment, and Review of Information (SUMARI) up to June 2022. We have completed that and are mapping that information as well. We will address each of your suggestions and ask for more time to complete this. 

Round 2

Reviewer 1 Report

NA

Author Response

Thank you for your time in reviewing our paper. The spelling and grammar issues were revised.

Reviewer 3 Report

1. A clearer separation in the discussion of results between human trials and animal models should be given

2. Some spell check required (line 81 misses "tract", some c and g are misspelled,...)

Author Response

  1. A clearer separation in the discussion of results between human trials and animal models should be given

On lines 248-250 we added a human trial study after the animal study and specifically indicated animal versus human trial. 

     2.Some spell check required (line 81 misses "tract", some c and g are misspelled,...)

We revised spelling (line 81) and grammar issues as you suggested. 

Thank you for your review.